# Probiotics Dietary Supplementation for Modulating Endocrine and Fertility Microbiota Dysbiosis

**DOI:** 10.3390/nu12030757

**Published:** 2020-03-13

**Authors:** Ana López-Moreno, Margarita Aguilera

**Affiliations:** 1Department of Microbiology, Faculty of Pharmacy, University of Granada, Campus of Cartuja, 18071 Granada, Spain; 2Institute of Nutrition and Food Technology “José Mataix”, Center of Biomedical Research, University of Granada, 18016 Armilla, Granada, Spain; 3IBS: Instituto de Investigación Biosanitaria ibs., 18012 Granada, Spain

**Keywords:** probiotics, doses, microbiota, endocrine, fertility, dysbiosis

## Abstract

Human microbiota seems to play a key role in endocrine and reproductive systems. Fortunately, microbiota reproductive dysbiosis start to be treated by probiotics using typical species from genus *Lactobacillus*. This work presents the compiled and analysed results from the most up-to-date information from clinical trials regarding microbiota, fertility, probiotics and oral route administration, reviewing open access scientific documents. These studies analyse the clinical impact of probiotics administered on several endocrine disorders’ manifestations in women: mastitis; vaginal dysbiosis; pregnancy complication disorders; and polycystic ovary syndrome. In all cases, the clinical modulation achieved by probiotics was evaluated positively through the improvement of specific disease outcomes with the exception of the pregnancy disorders studies, where the sample sizes results were statistically insufficient. High amounts of studies were discarded because no data were provided on specific probiotic strains, doses, impact on the individual autochthon microbiota, or data regarding specific hormonal values modifications and endocrine regulation effects. However, most of the selected studies with probiotics contained no protocolised administration. Therefore, we consider that intervention studies with probiotics might allocate the focus, not only in obtaining a final outcome, but in how to personalise the administration according to the disorder to be palliated.

## 1. Introduction

### 1.1. Human Microbiota and Reproductive Impact

The microbial communities that live in the human body, called microbiota, have an important role in the eubiosis status of each individual [1]. A change in its natural state (dysbiosis) have been shown to contribute to endocrine and metabolic disorders such as bacterial vaginosis [2], polycystic ovary syndrome (PCOS) [3], endometrial hyperplasia and endometriosis [4], and several comorbidities like obesity [5] and diabetes [6].

The vaginal microbiota is particular to each woman, and its role in many facets of reproductive health is increasingly substantiated [1,7]. Among all microbial communities that make up the vaginal microbiota, *Lactobacillus* spp. are the main representative genus for establishing a healthy community [8]. The reasoning behind this physiological impact might be that gastrointestinal microbiota also has a very important active metabolic role in the regulation of the hormonal physiological axis. The GIT microbiota may regulate circulating oestrogens by the secretion of β-glucuronidase. This enzyme deconjugates oestrogens into their active forms, allowing them to bind their receptors. However, if microbial dysbiosis occurs, decreasing bacterial diversity results in a deconjugation decrease and a reduction of circulating oestrogens. This alteration can contribute to the development of hormonal-dependent pathologies [4,9].

### 1.2. Dietary Nutritional Supplements: Probiotics and Nutraceuticals

Probiotics, according to the International Life Sciences Institute (ILSI), WHO, and the International Scientific Association of Probiotics and Prebiotics (ISAPP), are defined as “live microorganisms, which when administered in adequate amounts confer a health benefit on the host” [10].

Nutraceuticals are defined as food product that provide therapeutic or physiological benefits beyond the basic nutritional needs and can be used as dietary supplements. Nutraceuticals are required to be safe and well tolerate, to exhibit less toxicity and secondary side effects compared to drugs used to treat similar symptoms [11,12,13]. Nutraceuticals and released bioactive compounds by probiotics can be used as therapeutic tools to modulate the human microbiota.

Probiotic strains are selected pseudo-empirically for clinical studies and also based on their previously proven beneficial effects, specific activity or function and safety aspects [14]. Administration of probiotics in clinical studies is also based on the demonstration of safety aspects and stronger probiotic activity [15]. Currently, Health Claims constitute a method for verification of demonstrable beneficial properties of probiotic products through compilation of data such as product characterisation, measurable beneficial effects, and links between the product and the exerted effect and further assessment [16].

Therefore, it is necessary to well identify the microorganisms used in the probiotic formula. There exist high throughput molecular tools (whole genome sequence) and chemotaxonomic techniques (MALDI-TOF) to identify microorganisms used in the chain food, and it is requested to fulfil, first of all, strict taxonomical criteria [17]. Moreover, once it is clarified which strains of probiotics should be recommended, the administration pattern, doses and periods of the intervention should be known before the intake [5].

### 1.3. Probiotics Administered in Fertility Dysbiosis

Today, bacterial vaginosis (BV) is one of the most common forms of vaginal dysbiosis and plays a role in gynaecological complications, such as spontaneous preterm birth, abortion, and endometritis [18,19]. BV can be treated by the restauration of vaginal microbiota environment using probiotics. Most commonly used probiotics for modulating fertility dysbiosis include *L. reuteri* RC-14, *L. fermentum*, *L. gasseri*, *L. rhamnosus*, *L. acidophilus*, *L. crispatus*, *L. casei*, *L. salivarius* [20,21,22,23]. Clinical trials have shown that these probiotics can play a positive role in restoring vaginal microecology and treating BV [24]. Probiotics could also avoid the abuse of antibiotics and further side effects.

### 1.4. Type of Administration and Site of Colonisation of Probiotics

There are different administration routes for probiotics: dermal, rectal, vaginal and oral, the latter being most common. However, for oral administration, probiotics are required to survive low pH of gut and gastric liquids in the intestine. Therefore, the survival of probiotics in the gastrointestinal tract is usually demonstrated by recovering strains from faecal samples [25,26]. It is also very important to guarantee the transfer to the final site of colonisation, in order to exert the expected clinical effect. Therefore, in the topic of research on fertility, probiotics could reach the vaginal niche by physical transfer or ascendant route, haematogenous route and lymphatic nodes transfer [27]. For the treatment of vulvovaginal infections, probiotics can be administered by the vaginal route in order to control the *Lactobacilli* recolonisation [25], as a result, these probiotics do not pass through the gastrointestinal tract protecting them from the acidic environment.

We identified this area of research as continuously in progress due to the fact of the growing interest in it by fertility clinicians and patients. Therefore, the main objective of this work was to search, review, extract and present the most up-to-date and relevant scientific evidence from the literature on probiotics and its effective modulating role in endocrine disorders related to fertility, using innovative literature methodologies and tools. This review will also improve and unify our knowledge about the administration of probiotics and their effects in this field of clinical research.

## 2. Materials and Methods

### 2.1. Search Strategy

Literature search and review of clinical studies administering specific probiotics strain formula to subjects suffering endocrine and fertility-related disorders were carried out between October 2019 and January 2020 under the stepwise procedure search. The following databases were used: MEDLINE/PubMed [28], Web of Science (Thomson Reuters Scientific), Scopus (Elsevier), and Cochrane Library [29] with search strategies following the search equation and search formula, adapted to each database’s tutorials. The combined search approach was performed as follow: (probiotic* AND infertility AND doses); (probiotic * AND microbiota AND fertility); (probiotic * AND microbiota AND infertility); (probiotic * AND “vaginal microbiota” AND infertility); (probiotic * AND endometriosis); (probiotic * AND endometriosis AND fertility); (probiotic * AND endometriosis AND infertility); (probiotic * AND “endometrial microbiota” AND infertility); (probiotic * AND endometrium AND infertility); (probiotic * AND endometrium AND fertility); (probiotic * AND microbiota AND “*vaginal administration”); (probiotic * AND microbiota AND “oral * administration” AND doses); (probiotic * AND ovules); (probiotic * AND reproduct * AND “oral * administration”); (probiotic * AND reproductive AND “*vaginal administration”); (probiotic * AND “Polycystic Ovary Syndrome” AND “oral * administration”); (probiotic * AND “Polycystic Ovary Syndrome”).

### 2.2. Eligibility Criteria

To be included in the study, there were four mandatory inclusion criteria: (1) it was published within the last fifteen years (i.e., between 2005 and 2020) and it specified the (2) probiotic strain used, (3) the dose and the (4) time/period of administration. The specific data on population, intervention, comparison and outcome criteria for inclusion are detailed in Table 1.

Exclusion criteria: non-open access documents; non-English-language manuscripts; documents containing no quantitative/biomarker-specific data, e.g., endpoint or outcome; systematic reviews; studies with data results from dysbiosis or not related with fertility; publications without accessibility to the results; publications such as comments, editorials or letters (grey literature).

Each identified eligible article was re-analysed by title and abstract, and the eligible articles were selected for full reading.

The first selection was done based on a term search through title and abstract screening, and the second selection was based on a full-text screening. The first reviewer, A.L.M., conducted the two screening phases, and M.A., a second reviewer supervised the selection. The raw literature search yielded a total of 719 clinical studies with probiotics used to treat metabolic diseases. Furthermore, studies were selected after full-text screening when they met the eligibility criteria. In cases of doubt, articles were also thoroughly analysed based on their full-text content. A PRISMA (preferred reporting items for literature search on the topic of interest [30]) flow diagram of the literature search summarises the selection of the studies consisted of two screening phases (Figure 1).

#### Data Analysis and Extraction

The following data were collected: characteristics of population included: sample size (*n* = number of subjects) in the intervention group, gender for clinical trials, specific disease; bacterial probiotic strains used; dose and/or pattern of administration; modification of the clinical parameters included: Nugent score or alterations in fertility-related parameters. Data presentation is summarised in Table 2. Further specifications and key results were retrieved from the CTs in order to visualize the relevance of the probiotics administered and the particular capacities of modulating fertility related dysbiosis.

### 2.3. Risk of Bias (Quality) Assessment

We (i.e., A.L.M.) independently assessed the risk of bias of the selected documents using the Cohrane collaboration’s methodology. A second reviewer (M.A.) was involved in this evaluation thoroughly. Risk of bias was tabulated for each study (Figure 2), and each item evaluated was classified as low risk (−), high risk (+) or unclear risk (?) (Figure 3), according to recommendations described in Chapter 8 of the *Cochrane Handbook of Systematic Reviews of Interventions* [28]. This analysis and correspondent figures were generated in RevMan 5.3.

## 3. Results

The present systematic review focuses on the selection of fully accessible data about probiotics used for modulating dysbiotic microbiota in fertility disorders. We found 222 studies in accordance with our inclusion criteria from a total of 719 documents. Only ten of these chosen studies were clinical trials, and the main results are presented in Table 2. The method applied for selecting the final documents guaranteed the quality of these studies. Moreover, the possibilities to assess the risk of biases of the CT designs, execution and outcomes increased the categorization of the applied quality standards. It gave added value to the evaluated CTs and allowed validating the reviewed results (Figure 2; Figure 3).

The probiotics doses and the duration of interventions varied from 1 × 10^6^ CFU/day to 3 × 10^10^ CFU/day and from 3 to 24 weeks in the selected clinical trials. None of the probiotics used triggered any safety concerns.

In regard to the disorders and symptoms treated by probiotics, we found 4 articles about women suffering mastitis due to the presence of various causes including hormonal imbalances. All studies showed a significant improvement in the clinical status of the patient groups that received specific strains of *Lactobacillus* [21,23,31,32]. In the case of PCOS patients, one study administered *Bifidobacterium lactis* V9 to 14 patients and showed positive modulation capacities of sex hormone levels [33]. Moreover, the controlled administration of probiotics during pregnancy, microbiota disorders allowed us to select only two clinical studies, but the sample size was insufficient to conclude specific modulation capacities [18,34]. Finally, for the treatment of vaginal dysbiosis, there was one study that proved the use of specific probiotics for restoring vaginal microbiota towards normal Nugent score [35] and one study that reduced successfully the recurrence of colonisation by pathogenic bacteria [36].

We further reviewed, extracted and highlighted the relevant information from the selected studies. Therefore, the study carried out by Zhang et al. [33] in PCOS patients revealed the relevance of determining the individual physiological status before administering the probiotics to be tested. Data determinations on microbiota compositional variability, key metabolites (SCFA, TC), gut–brain mediators (Ghrelin, PYY) and sex hormones levels were decisive for the outcome comparisons. These authors also postulated the potential mechanism undergoing the regulation of LH and LH/FSH hormones levels by *Bifidobacterium lactis* V9 supplementation.

The evaluation of the bacterial probiotics *Lactobacillus rhamnosus* GR-1 and *Lactobacillus reuteri* RC-14/15 and their modulation capacities on pregnancy associated problems, such as avoiding preterm birth, were also studied in relation to the individual vaginal dysbiosis status [18,34]. Both studies administered differential doses of probiotics (1 × 10^9^ CFU/day over 4 weeks and >2 × 10^6^ CFU/day during 6–12 weeks); however, the results demonstrated the prevention of premature delivery comparing to placebo groups. In any case, the final sample size adhered to the probiotics treatment, and the lack of further experimental assays did not allow for the obtention of robust statistical significance for the probiotic intent-to-treat effects. For similar statistical reasons, there was no possibility to evaluate the differential patterns of microbiota colonisation in preterm infants when administering *Saccharomyces boulardii* [37].

Probiotics have been used for avoiding mastitis symptoms. At this point, four studies had fulfilled the selection criteria and were reviewed thoroughly. The clinical trial designed by de Andrés et al. [23] applied omics technologies to describe the interindividual variability of gene expression changes in somatic cells and blood leukocytes by the action of *Lactobacillus salivarius* PS2. The results demonstrated its modulating role in inflammatory and cell-grow only in the breast milk somatic cells. The results indicated that clinical interventions would take advantages from personalized use of probiotics therapy due to the individual response variability linked to the complexity of gene expressions. The study from Jiménez et al. [31] administered *Lactobacillus salivarius* CECT5713 and *Lactobacillus gasseri* CECT5714 to the probiotic group versus placebo to the control group. For later analysis of the progression of mastitis, they counted and identified bacteria in the milk samples, finding no clinical signs of mastitis in the probiotic group at day 14, but mastitis persisted on the placebo group. Also, at day 30 probiotic group showed a lower staphylococcal count than the control group (10^2^–10^3^ CFU/mL; 10^4^–10^5^ CFU/mL, respectively). Both probiotic strains administered together appeared to be efficient for treating infectious mastitis during lactation. In the case of Arroyo et al. [21], they administered *L. fermentum* CECT5716 and *L. salivarius* CECT5713 separately to two probiotic groups and showed at day 21 almost 1 log10 CFU/mL staphylococcal lower count in probiotic groups than antibiotic group. Also, probiotics can be administered during pregnancy for the mastitis prevention. Fernández et al. [32] administered *L. salivarius* PS2 during late pregnancy, and only 25% of women in the probiotic group developed the disease in contrast with 57% of women in the control group, being efficient to prevent infectious mastitis in a susceptible population.

Vaginal dysbiosis interventional studies with probiotics also showed similar clinical improvements, such as less vulvovaginitis recurrences, but without modifying cure rates. Moreover, oral probiotics supplementations decreased Nugent score [35,36]. Oral *Lactobacillus* probiotics tested were able to successfully resist the conditions of gastric tract for colonizing temporarily the vaginal area. In addition, they showed high level of safety according to the administered doses and treatment patterns.

In addition, nine related studies were identified and selected from animal clinical studies (Appendix A) and one systematic review (Appendix A) from human clinical data that could both support several hypotheses of probiotics colonisation, transfer route and clinical effects. These studies could also give some added value for comparing probiotic capacities of modulation during the intervention. Transversally, we found three interesting articles about sperm quality with three different outpoints: a decrease in the quality [38], zero negative effects [39] and an increase in the quality [40] with different probiotics administration pathways. The systematic review selected for comparison of the results analysed three articles focused on treatments of vaginal dysbiosis, and the achievement of reduction of colonisation by pathogenic bacteria [20].

## 4. Discussion

In recent years, the relevance of probiotics and their beneficial effects through modulating microbiota in several disorders have been extended within the clinicians. Moreover, probiotic supplements and their active compounds formulated as nutraceuticals have become better accepted for general consumers and potential patients [24,41,42]. Probiotics have also shown scientific evidence for their positive role in fertility disorders and for hormonal imbalances. This could be due to the increasing evidence of abundance and impact of microorganisms on the reproductive tract, so it is pertinent to deeply explore the use of probiotics in the context of endocrine and reproductive health [1].

There are an increasing number of research studies proving the beneficial effects of probiotic oral supplementation. Moreover, at this time, there is unequivocal evidence of a direct beneficial effect on reproductive health clinical outcomes. All the same, oral administration of *Lactobacillus rhamnosus* GR-1 and *Lactobacillus fermentum* RC-14 have been shown to restore healthy vaginal microbiota in up to 82% of women with previous vaginal dysbiosis, specifically an increase in *Lactobacillus* species [43]. In addition, a recent study examined the relationship between the composition of the endometrial microbiota and infertility, showing 30% to 71% of infertile women have endometriosis [44]. Interestingly, Moreno et al. [45] found also a correlation between adverse pregnancy outcomes and endometrial microbiota’s lower colonisation in *Lactobacillus* species, concluding by associating this fact with the negative effects, poor reproductive outcomes, implantation failure and pregnancy loss [46,47,48].

The presence of bacterial microbiota in the placenta during pregnancy is not well supported. There are still controversial studies regarding human uterine and placenta microbiome and the impact on pregnancy and foetus [49,50,51,52]. Moreover, establishment and maintenance of placental integrity and function are critical to foetal growth, development and survival [53]. Also, the presence of ruptured membranes causes bacterial infections, 70% on average, which elicit a maternal inflammatory response [54]. Before the omics technologies era, the sterile human foetal environment (placenta, foetus and amniotic fluid) and the acquisition of the microbiota during and after delivery was an accepted dogma [49].

Recently, studies using sequencing-based methods suggested the presence of bacterial communities in the placenta. Aagaard et al. [51] characterized a unique placental microbiome niche composed of non-pathogenic commensal microbiota. Prince et al. [52] found a relationship among the alterations of the placental microbiota with the severity of chorioamnionitis. Both studies [51,52] showed a lower abundance of the microbial community in the placenta analyses. Conversely, recent studies identifying this low abundance do so with potential false-positive results, due to the fact of an insufficient detection limit using sequencing-based methods [49] or reagent and laboratory contamination [55,56].

As previously commented, formula supplementations containing probiotic strains are increasing as dietary and therapeutic products [57]. Regardless of the scarce data on probiotic strains, many authors only report on genus and species data [58]. However, it is well-known that the biological effects of probiotics are strain-specific dependent [59]. The effects of one strain cannot be extrapolated to another strain. Accordingly, the specific benefits of the probiotic strain highlight the need to include a complete identification of the probiotic to the strain level [60]. In any cases, it is required to carry out more studies and test the beneficial impact of probiotics, because in many cases, there are only one clinical trial that supported their claims [58]. In addition, many other parameters need to be harmonized, such as the pattern of administration or target population, since it has been seen that the beneficial effect of a probiotic in a population may not be adequate for another population, even causing potential adverse effects [57,61].

In summary, the collection of relevant results pursued in the present review were the verification of the availability of complete key clinical data during the accomplishment of probiotic fertility dysbiosis interventions regarding the following relevant aspects: selection of probiotic strain, doses, administration pattern and key endpoints modulation capacities. Interestingly, from a total of 719 studies initially gathered, only ten clinical trials fulfilled the inclusion criteria (Figure 1). In contrast, there were an enormous amount of clinical studies without these specifications (see Appendix A).

Most of the probiotics used were *Lactobacillus* species, due to the fact of their natural predominance in these colonizing sites [62] and other manufacturing and technological concerns, e.g., anaerobiosis [63]. Similarly, specific probiotic strains of *Lactobacillus* were also selected for vaginal dysbiosis studies in human clinical trials (Table 2) and animal studies (Appendix A). Some species of *Lactobacillus*, *L. acidophilus*, *L. crispatus*, *L. plantarum*, *L. fermentum* and *L. gasseri* makeup the predominant normal microbiota of the genitourinary and gastrointestinal (GI) tract of healthy individuals, and its effectiveness in maintaining the normal pH of the vagina and preventing genital infections has been well demonstrated [64,65]. Within the genus *Lactobacillus*, as described above, *L. gasseri* has been widely used as a probiotic. In the study of Itoh et al. [66], the administration of *L. gasseri* OLL2809 heat-killed achieved the suppression of the development of endometriosis in mice. The selection of this strain was based on its immunostimulatory activity [67,68]. Qin et al. [69] compared the effects of *Lactobacillus rhamnosus* CICC6141 and *Lactobacillus casei* BL23 on *Danio rerio* and, either separately or in combination, they found stimulation in parameters of the reproductive process. The process of probiotic selection was done according to the level of adhesion to the epithelial gut capacities [70].

Besides, *Lactobacillus* have been used in studies of mastitis or colonisation of the mammary gland (MG), as in the Treven et al. [71] study, in which *L. gasseri* K7 and *L. rhamnosus* GG can modulate the bacterial composition of the mammary gland. These bacteria were selected by the previous demonstration of modulation of MG colonisation capacities and, consequently, colonisation of the infant’s gut [72,73]. However, in the case of human clinical trials of mastitis, the predominant probiotic strains used are *Lactobacillus salivarius* PS2, *L. salivarius* CECT5713, *L. gasseri* CECT5714, *L. fermentum* CECT5716 [21,22,23,31,32], also the doses and period of administration were quite similar (3–4 weeks), with the exception of Fernández et al. [32] in which study’s period of administration lasted about 8 weeks. In all studies, the strains administered seem to be an efficient method for the treatment of mastitis and other hormonal imbalances triggering infections, as alternatives to antibiotics. Finally, probiotics of the genus *Lactobacillus* have also been used to treat vaginal dysbiosis. Specifically, *L. rhamnosus* GR-1, *L. reuteri* RC-14, *L. casei* LCR35, *L. fermentum* RC-14, *L. fermentum* RC-15 and *L. fermentum* RC-16 [20,35,36] strains, thus improving the microbiota vaginal to normal or intermediate Nugent scores and reducing recurrence of pathogen colonisation.

Exceptionally, despite the common use of the genera *Lactobacillus*, also in animal studies, dos Santos et al. [38] administered *Bacillus subtilis* to roosters, due to the fact of its use as a dietary supplement to prevent gastrointestinal disorders and enhance growth performance [74]; however, no effects or lack of conclusive information were found regarding the fertility assays. Conversely, few species of *Bacillus* are considered suitable for commercialized oral dietary supplements. They have been commonly approved for their history of safe use (*Bacillus subtilis* (DSM10), *Bacillus clausii* (DSM 8716), *Bacillus coagulans* (DSM 1), *Bacillus amyloliquefaciens* (DSM 7) [75]. Moreover, Valcarce et al. [40] studied in animal models the effects of *Lactobacillus rhamnosus* CECT8361 and *Bifidobacterium longum* CECT7347 on sperm quality finding an improvement of the parameters, these probiotics were selected because of its antioxidant and anti-inflammatory activities [40,76]. On the other hand, probiotics of the genus *Bifidobacterium* have been used for the treatment of PCOS, specifically the strain *Bifidobacterium lactis* V9, previously tested by Zhang et al. [33] and selected for exhibiting excellent probiotic characteristics.

Some studies have been conducted to prove the capacity of orally administered Lactobacilli to be transferred from the intestinal tract to the vagina and positively colonize and influence vaginal health [20,35]. Comparatively, despite vaginal administration allowed a direct and targeted colonisation action of the probiotics for restoring unhealthy vaginal microbiota, other many studies defended the thesis that oral administration is more effective against bacterial vaginosis, and less aggressive for established microbiota, and, in summary, it constitutes a natural route of colonisation [77].

Finally, most of the clinical trials reviewed reported on the safety aspects of the prescribed probiotics during the interventions. None of the studies showed data on side or adverse effects triggered by the probiotics administered. All probiotics used should be innocuous by history of safe use and proven safety features [78] for commercial products marketed [79]. The commercial probiotic species used should have the qualifications QPS (qualified presumption of safety) [80] or GRAS (generally recognized as safe) [81].

## 5. Conclusions

The clinical modulation achieved by specific doses of oral administered probiotics was evaluated positively through the improvement of the specific hormonal- and fertility-related disease outcomes with the exception of two pregnancy disorder studies. However, most of the selected studies contained no data harmonised on the probiotic administration, and it made it difficult to clinically protocolised the probiotic modulation capacities. Therefore, we consider that future interventional studies with probiotics might allocate the focus, not only in obtaining a final outcome, but in how to personalise the administration according to the disorder to be palliated.

## Figures and Tables

**Figure 1 nutrients-12-00757-f001:**
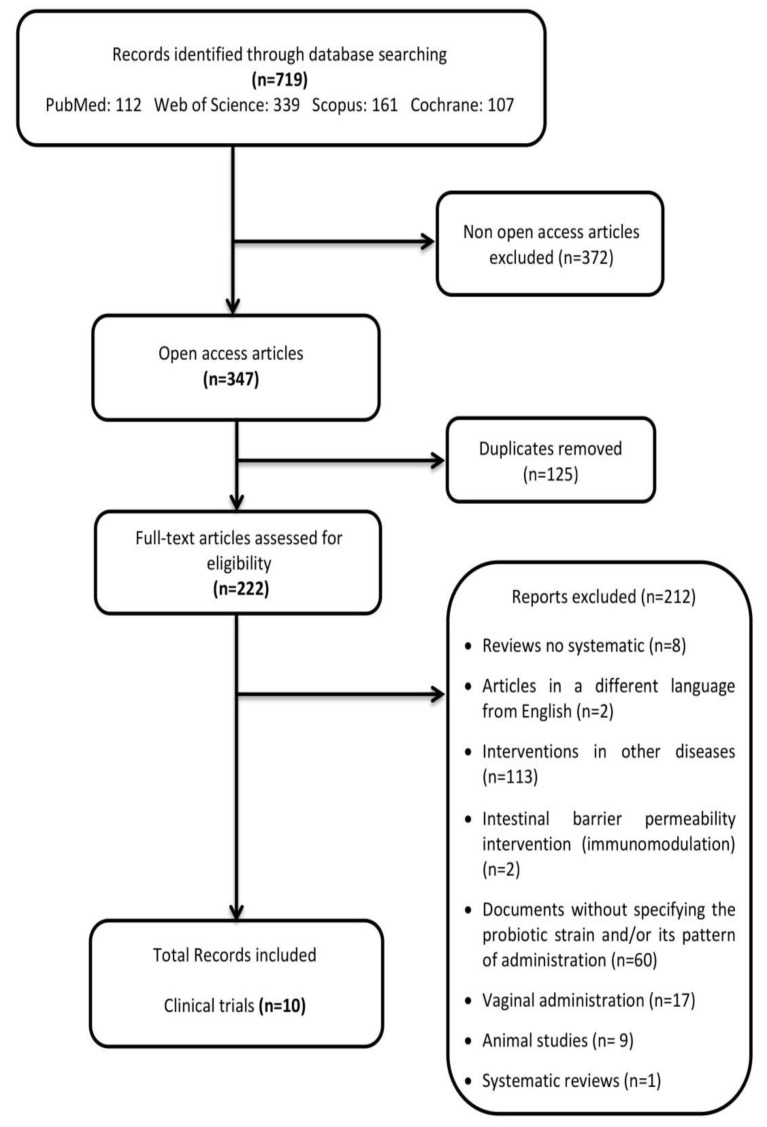
PRISMA flow diagram: preferred reporting items for literature search.

**Figure 2 nutrients-12-00757-f002:**
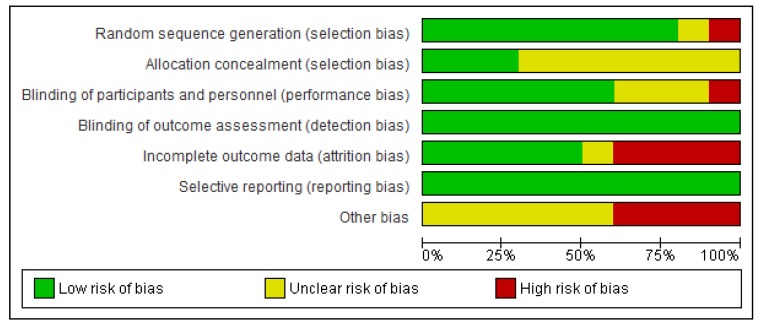
Risk of bias graph: review authors’ judgments about each item presented as percentages across all included studies.

**Figure 3 nutrients-12-00757-f003:**
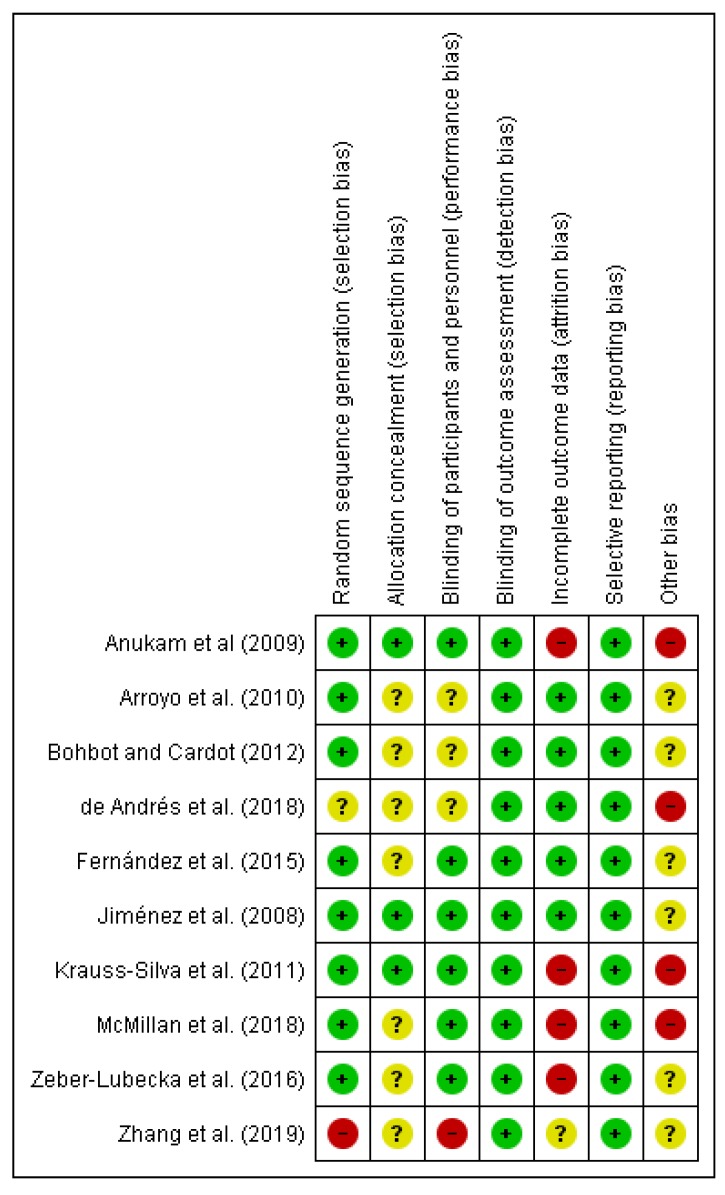
Risk of bias summary: review authors’ judgments about each risk of bias item for each included study.

**Table 1 nutrients-12-00757-t001:** PICO (population, intervention, comparison, outcome) criteria for inclusion of studies.

Parameters	Inclusion Criteria
Population	Human
Intervention	Probiotic strains and doses
Comparison	Oral probiotics versus placebo
Outcome	Improvement on parameters of fertility
Setting	Clinical trials (CTs)

**Table 2 nutrients-12-00757-t002:** Summary of the effects of probiotics on human fertility disorders and their clinical outcomes.

Reference	Population Characteristics Size (*n*)	Probiotic Strain	Doses and Administration Pattern	Period of Intervention (Weeks)	Disease	Clinical Parameters Variability
Zhang et al. [33]	14 PCOS patients	*Bifidobacterium lactis* V9	1 × 10^6^ CFU/day	10	PCOS	The study showed a potential mechanism by which the probiotic *Bifidobacterium lactis* V9 modulates sex hormone levels in patients with PCOS through the gut–brain axis.
McMillan et al. [34]	38 Pregnant women <36 weeks	*Lactobacillus rhamnosus* GR-1 and *Lactobacillus reuteri* RC-14	1 × 10^9^ CFU/day	4	Pregnancy associated disorders	Women in the placebo group were more likely to give birth preterm. However, the sample size that finished the study was not large enough to detect significant differences.
Krauss-Silva et al. [18]	4204 Pregnant women <20 weeks	*Lactobacillus rhamnosus* GR-1 and *Lactobacillus reuteri* RC-15	>2 × 10^6^ CFU/day	6–12	Pregnancy associated disorders	The efficacy of probiotics tested to avoid spontaneous premature delivery cannot be statistically estimated because the study sample was insufficient.
de Andrés et al. [23]	31 Women (23 with mastitis)	*Lactobacillus salivarius* PS2	3 × 10^9^–3 × 10^10^ CFU/day	3	Mastitis	The results proved the involvement of modulation of inflammatory and cell-growth related pathways and genes in the somatic cells following the intake of *L. salivarius* PS2.
Jiménez et al. [31]	20 Women with mastitis	*Lactobacillus salivarius* CECT5713 and *Lactobacillus gasseri* CECT5714	1 × 10^10^ CFU/day	4	Mastitis	Both probiotics appears to be an efficient alternative for the treatment of infectious mastitis during lactation.
Arroyo et al. [21]	352 Women with mastitis	*Lactobacillus fermentum* CECT5716 *Lactobacillus salivarius* CECT5713	1 × 10^9^ CFU/day	3	Mastitis	*L. fermentum* CECT5716 or *L. salivarius* CECT5713 seem to be an efficient alternative to antibiotics for the treatment of infectious mastitis during lactation.
Fernández et al. [32]	108 Healthy pregnant women	*Lactobacillus salivarius* PS2	1 × 10^9^ CFU/day	~8	Mastitis—Pregnancy	Administration of *L. salivarius* PS2 during late pregnancy appears to be an efficient method to prevent infectious mastitis in a susceptible population.
Zeber-Lubecka et al. [37]	39 Preterm infants	*Saccharomyces boulardii*	2 × 10^9^ CFU/day	6	Microbiota dysbiosis	There were no statistical differences between babies supplemented with probiotic and without probiotic.
Anukam et al. [36]	59 Women with vaginal dysbiosis	Fluconazol, *Lactobacillus rhamnosus* GR-1 y *Lactobacillus reuteri* RC-14	5 × 10^9^ CFU/day	24	Vaginal dysbiosis	Probiotics did not affect the cure rate but did lead to fewer vulvovaginitis recurrences with its long-term use.
Bohbot and Cardot [35]	20 Healthy women	*Lactobacillus casei* variety *rhamnosus* (LCR35)	Group 1: 1 × 10^8^ CFU/day Group 2: 2 × 10^8^ CFU/day	4	Vaginal dysbiosis	Probiotic decreased the Nugent score in both groups, but it was slightly more significant in group 2.

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
