# Peer review of "Probiotics Dietary Supplementation for Modulating Endocrine and Fertility Microbiota Dysbiosis"

_nutrients, 2020, doi:10.3390/nu12030757_

Round 1

Reviewer 1 Report

A very interesting review with welcome observations. Too much is spent explaining how the review was conducted instead of reviewing the literature. The way of presenting the data is more suitable for a more statistical journal than an expertise focused journal. Please, reconsider changing the title for a more appropriate to avoid confusing readers. The use of English language is bit ridged and the use of a native English speaker to proofread the manuscript. Also make sure to use the correct references.

R13 Please, refrase: “The human microbiota plays a key role in hormonal and reproductive system.”

R14 “Particularly, vaginal microbiota has shown a specific and individual colonizing pattern, and…”

R16 “Today, bacterial vaginosis…”  “… most common form…”

R18 “Fortunately, …”

R19 “…treated by probiotics using typically species from genus Lactobacillus. “

R22 “, …reviewing open access…”

R22-23 “These studies analyse the…”

R27 “ …where the sample…

R28-29 “High amount of studies were discarded because no data was available on specific…”

R46 “Lactobacillus spp. are the main representative genus…”

R48 Please, refer to “gastrointestinal microbiota” rather than “gut microbiota”

R56-58 Please, open up the term ILS and add the short term ISAPP. Also do use the correct reference for the statement: Hill, C. et al. Nat. Rev. Gastroenterol. Hepatol. 11, 506–514 (2014)

R59-60 ”…and probiotics that may also release bioactive compounds.”

R62 ”…needs and can be…”

R62-63 ”Nutraceuticals are required to be safe… … to exhibit toxicity and and secondary side effects compared to drugs.”

R65 ”Probiotic strains are selected pseudo-emperically for clinical studies and also based on…”

R70 ”…compilation of data, such as…”

R79-80 Please check the use of commas: ”Today bacterial… ….dysbiosis and ….complications, such as…”

R82 What is ment by common probiotics?  Most commonly used probiotics for humans or most commonly used for altering fertility dysbiosis? Please specify for clearance.

R88 ”…. different administration routes for…”

R89  Nutraceuticals are not under prescription :”…, the later being most common.”

R90-91 ”… probiotics are required to survive low pH of the gut and gastric liquids in the intestine. Therefore, the survival of probiotics in the gastrointestinal … … by recovering strains from….”

R108 ”Literature search and review…”

R128-129 Avoid repetition by combining rows 128-129 with rows 125-127. Also ”Specidic data on population, intervention, … … are detailed in Table 1.”

R130 -134 Unclear what is meant. Is manuscript in English a exclusion criteria?

R135 Spelling error in word ”title”

R141 ”… with probiotics used treating metabolic …”

R144 and R160  Replace ”exhaustively ” with ”thoroughly”

R158 Please, keep abbreviations in line: use ALM or A.L.M but not both.

R161 Why the question mark after ”unclear”?

R207 Please, specify the strains of L. fermentum and L. rhamnosus.

R211 In vitro in italics

R216-217 Please, discuss the microbial contact during pregnancy, uterine microbiota and bacterial transmission through the placental barrier.

R267 Please, specify the strain of Bacillus subtilis and check it has a probiotic status before discussing the strain under the topic of Probiotics.  Also, animal feed supplements are regarded as complimentary feeds, not dietary supplements.

R276 Capitalize ”Lactobacilli”

Author Response

Authors Corrections

Journal Nutrients (ISSN 2072-6643) Manuscript ID nutrients-731602

Reviewer 1

Reviewer comment

  • Authors corrections

A very interesting review with welcome observations.  

  • THANK YOU VERY MUCH FOR YOUR VALUABLE COMMENTS

Too much is spent explaining how the review was conducted instead of reviewing the literature.

  • Thank you for your valuable suggestion. We have built and inserted new paragraphs in order to highlight the most important achievements from each selected studies.

- Please see lines: 153-156 and 200-245

The way of presenting the data is more suitable for a more statistical journal than an expertise focused journal.

  • We partially agree with this constructive comment; however we consider this methodological approach useful for categorizing the selected documents.
  • Please also note that this approach involved a double check of key parameters defining the differences encountered during the Clinical Trials accomplishment (design, methods and outcomes). We consider that this analysis could give a kind of validation of the results obtained.

-Please see lines: 181-185

Please, reconsider changing the title for a more appropriate to avoid confusing readers.

  • Title has been changed: Probiotics dietary supplementation for modulating endocrine and fertility microbiota dysbiosis

The use of English language is bit ridged and the use of a native English speaker to proofread the manuscript.

  • Thank you for your valuable suggestion. It has been DONE along with all corrections suggested.

Also make sure to use the correct references.

  • Thank you for your valuable suggestion. It has been DONE and we have inserted new references according to new paragraphs built.

R13 Please, refrase: “The human microbiota plays a key role in hormonal and reproductive system.”

  • Thank you for your valuable suggestion. It has been DONE and inserted in the correspondent line as follow:

“Human microbiota seems to play a key role in endocrine and reproductive systems”.

R14 “Particularly, vaginal microbiota has shown a specific and individual colonizing pattern, and…”

  • Thank you for your valuable suggestion/correction. It has been DONE and inserted in the correspondent line.

R16 “Today, bacterial vaginosis…”  “… most common form…”

  • Thank you for your valuable suggestion/correction. It has been DONE and inserted in the correspondent line.

R18 “Fortunately, …”

  • Thank you for your valuable suggestion/correction. It has been DONE and inserted in the correspondent line.

R19 “…treated by probiotics using typically species from genus Lactobacillus. “

  • Thank you for your valuable suggestion/correction. It has been DONE and inserted in the correspondent line.

R22 “, …reviewing open access…”

  • Thank you for your valuable suggestion/correction. It has been DONE and inserted in the correspondent line.

R22-23 “These studies analyse the…”

  • Thank you for your valuable suggestion/correction. It has been DONE and inserted in the correspondent line.

R27 “ …where the sample…

  • Thank you for your valuable suggestion/correction. It has been DONE and inserted in the correspondent line.

R28-29 “High amount of studies were discarded because no data was available on specific…”

  • Thank you for your valuable suggestion/correction. It has been DONE and inserted in the correspondent line.

R46 “Lactobacillus spp. are the main representative genus…”

  • Thank you for your valuable suggestion/correction. It has been DONE and inserted in the correspondent line.

R48 Please, refer to “gastrointestinal microbiota” rather than “gut microbiota”

  • Thank you for your valuable suggestion/correction. It has been DONE and inserted in the correspondent line.

R56-58 Please, open up the term ILS and add the short term ISAPP. Also do use the correct reference for the statement: Hill, C. et al. Nat. Rev. Gastroenterol. Hepatol. 11, 506–514 (2014)

  • Thank you for your valuable suggestion/correction. It has been DONE and inserted in the correspondent line. Moreover, Hill et al reference has been inserted as requested.

R59-60 ”…and probiotics that may also release bioactive compounds.”

  • Thank you for your valuable suggestion/correction. It has been DONE and inserted in the correspondent line.

R62 ”…needs and can be…”

  • Thank you for your valuable suggestion/correction. It has been DONE and inserted in the correspondent line.

R62-63 ”Nutraceuticals are required to be safe… … to exhibit toxicity and and secondary side effects compared to drugs.”

  • Thank you for your valuable suggestion/correction. It has been DONE and inserted in the correspondent line.

R65 ”Probiotic strains are selected pseudo-emperically for clinical studies and also based on…”

  • Thank you for your valuable suggestion/correction. It has been DONE and inserted in the correspondent line.

R70 ”…compilation of data, such as…”

  • Thank you for your valuable suggestion/correction. It has been DONE and inserted in the correspondent line.

R79-80 Please check the use of commas: ”Today bacterial… ….dysbiosis and ….complications, such as…”

  • Thank you for your valuable suggestion/correction. It has been DONE and inserted in the correspondent line.

R82 What is ment by common probiotics?  Most commonly used probiotics for humans or most commonly used for altering fertility dysbiosis? Please specify for clearance.

  • Thank you for your valuable suggestion. It has been DONE and specify that data referred to probiotics most commonly used for altering fertility dysbiosis.

R88 ”…. different administration routes for…”

  • Thank you for your valuable suggestion/correction. It has been DONE and inserted in the correspondent line.

R89  Nutraceuticals are not under prescription :”…, the later being most common.”

  • Thank you for your valuable suggestion/correction. It has been DONE and inserted in the correspondent line.

R90-91 ”… probiotics are required to survive low pH of the gut and gastric liquids in the intestine. Therefore, the survival of probiotics in the gastrointestinal … … by recovering strains from….”

  • Thank you for your valuable suggestion/correction. It has been DONE and inserted in the correspondent line.

R108 ”Literature search and review…”

  • Thank you for your valuable suggestion/correction. It has been DONE and inserted in the correspondent line.

R128-129 Avoid repetition by combining rows 128-129 with rows 125-127. Also ”Specidic data on population, intervention, … … are detailed in Table 1.”

  • Thank you for your valuable suggestion/correction. It has been DONE and inserted in the correspondent line.

R130 -134 Unclear what is meant. Is manuscript in English a exclusion criteria?

  • Thank you for your valuable suggestion/correction. Has been modified for better comprehension.

R135 Spelling error in word ”title”

  • Thank you for your valuable suggestion/correction. It has been DONE and inserted in the correspondent line.

R141 ”… with probiotics used treating metabolic …”

  • Thank you for your valuable suggestion/correction. It has been DONE and inserted in the correspondent line.

R144 and R160  Replace ”exhaustively ” with ”thoroughly”

  • Thank you for your valuable suggestion/correction. It has been DONE and inserted in the correspondent line.

R158 Please, keep abbreviations in line: use ALM or A.L.M but not both.

  • Thank you for your valuable suggestion/correction. It has been DONE and inserted in the correspondent line.

R161 Why the question mark after ”unclear”?

  • Thank you for your valuable suggestion/correction. Has been modified for better comprehension.

R207 Please, specify the strains of L. fermentum and L. rhamnosus.

  • Thank you for your valuable suggestion/correction. It has been DONE and inserted in the correspondent line.

R211 In vitro in italics

  • Thank you for your valuable suggestion/correction. It has been DONE and inserted in the correspondent line.

R216-217 Please, discuss the microbial contact during pregnancy, uterine microbiota and bacterial transmission through the placental barrier.

  • Thank you for your valuable suggestion. It has been DONE a comment for discussing this issue and it has been inserted in the correspondent lines.

Please see lines: 274-289

R267 Please, specify the strain of Bacillus subtilis and check it has a probiotic status before discussing the strain under the topic of Probiotics.  Also, animal feed supplements are regarded as complimentary feeds, not dietary supplements.

  • Thank you for your valuable suggestion. It has been DONE a comment for clarifying this issue and complement with some further data.

Please see lines: 340-347

R276 Capitalize ”Lactobacilli”

  • Thank you for your valuable suggestion/correction. It has been DONE and inserted in the correspondent line.

Reviewer 2 Report

In the present review, using innovative literature methodologies and tools, the authors analyzed selected papers from literature on probiotics and its effective modulating role in endocrine disorders related to fertility. The authors highlight a positive clinical modulation, achieved by specific doses of oral administered probiotics, evaluated through the increases of the specific hormones and fertility. Because the difficulty to potocolized the modulation capacities of probiotic the authors suggest, in the future studies, the necessity to harmonized the data on probiotics administration.

I think that the paper can be accepted for publication after resolving the following minor points

Line 10:  substitute “orresponding” with “corresponding”

Line 58-60  from “Another……… to compound “please rewrite more clearly 

Line 110 October (specify year)

Line 135 change “tittle” with “title”

Line 190,191,194  “tree” should be “three”

Line 211 “In Vitro” should be write in Italic

Line 257 “use” should be “used”

Line 262 “as alternatively” should be “as alternative”

Line 288 please write in full QPS and GRAS

Author Response

Authors Corrections

Journal Nutrients (ISSN 2072-6643) Manuscript ID nutrients-731602

Reviewer 2

Reviewer comment

  • Authors corrections

Comments and Suggestions for Authors

In the present review, using innovative literature methodologies and tools, the authors analyzed selected papers from literature on probiotics and its effective modulating role in endocrine disorders related to fertility. The authors highlight a positive clinical modulation, achieved by specific doses of oral administered probiotics, evaluated through the increases of the specific hormones and fertility. Because the difficulty to protocolized the modulation capacities of probiotic the authors suggest, in the future studies, the necessity to harmonized the data on probiotics administration. I think that the paper can be accepted for publication after resolving the following minor points

THANK YOU VERY MUCH FOR YOUR VALUABLE COMMENTS

 Line 10:  substitute “orresponding” with “corresponding”

  • Thank you for your valuable suggestion/correction. It has been DONE and inserted in the correspondent line.

Line 58-60  from “Another……… to compound “please rewrite more clearly 

  • Thank you for your valuable suggestion. It has been DONE a new paragraph for clarification the information.

Please see lines:

Line 110 October (specify year)

  • Thank you for your valuable suggestion/correction. It has been DONE and inserted in the correspondent line.

Line 135 change “tittle” with “title”

  • Thank you for your valuable suggestion/correction. It has been DONE and inserted in the correspondent line.

Line 190,191,194  “tree” should be “three”

  • Thank you for your valuable suggestion/correction. It has been DONE and inserted in the correspondent line.

Line 211 “In Vitro” should be write in Italic

  • Thank you for your valuable suggestion/correction. It has been DONE and inserted in the correspondent line.

Line 257 “use” should be “used”

  • Thank you for your valuable suggestion/correction. It has been DONE and inserted in the correspondent line.

Line 262 “as alternatively” should be “as alternative”

  • Thank you for your valuable suggestion/correction. It has been DONE and inserted in the correspondent line.

Line 288 please write in full QPS and GRAS

  • Thank you for your valuable suggestion/correction. It has been DONE and inserted in the correspondent line.

Round 2

Reviewer 1 Report

A very interesting review with welcome observations. A nice wrap up of the study.